# Development and Verification Experiment of In-Situ Friction Experiment Device for Simulating UV Irradiation in Space

**DOI:** 10.3390/ma15062063

**Published:** 2022-03-11

**Authors:** Aobo Wei, Qian Liu, Guozheng Ma, Wenbo Yu, Jiadong Shi, Yunfan Liu, Cuihong Han, Zhen Li, Haidou Wang, Guolu Li

**Affiliations:** 1School of Materials Science and Engineering, Hebei University of Technology, Tianjin 300130, China; weiaobo0625@163.com (A.W.); jiadong1207@126.com (J.S.); hanmutou@163.com (C.H.); 2National Key Laboratory for Remanufacturing, Army Academy of Armored Forces, Beijing 100072, China; 15120029165@163.com (G.M.); liuyf0016@163.com (Y.L.); whaidou2021@163.com (H.W.); 3School of Engineering and Technology, China University of Geosciences (Beijing), Beijing 100083, China; 13468552625@163.com; 4School of Mechanical, Electronic and Control Engineering, Beijing Jiaotong University, Beijing 100044, China; 5School of Mechanical Engineering, Shanghai Jiao Tong University, Shanghai 200240, China; lizhen2019@sjtu.edu.cn; 6National Engineering Research Center for Remanufacturing, Army Academy of Armored Forces, Beijing 100072, China

**Keywords:** ultraviolet irradiation, space, experimental apparatus, experimental verification, polymer

## Abstract

In order to explore the influence of space ultraviolet radiation on spacecraft lubricating materials, an in-situ friction experimental device simulating space ultraviolet radiation was developed in the laboratory, and the experimental verification was carried out. This paper firstly introduced the design index, structure and working principle of the space ultraviolet irradiation simulation device, and then calibrated and tested the parameters of the whole device, and also conducted a virtual operation of the device’s operation effect by simulation software, and the results showed that it met the design index. Finally, the validation tested of the ultraviolet irradiated in-situ friction experimental device were described in detail. By using the device to irradiate the samples, it was found that the in-situ ultraviolet irradiation device could achieve the expected irradiation effect, and the irradiation would lead to changes in the surface structure and properties of the PTFE material, while also achieving the need for in-situ spatial friction property testing of the material, providing favorable conditions for future testing.

## 1. Introduction

Space equipment is subject to intense solar irradiation during orbit [1]. The spectrum of solar radiation includes electromagnetic waves with various wavelengths, ranging from 10^−14^ m for γ rays to 10^4^ m for radio waves. The solar radiation energy is very large, which can be described quantitatively by solar constant (S0). It is defined as the total radiant energy per unit area of the outer edge of the atmosphere perpendicular to the solar rays at the mean distance between the sun and the earth within a unit time, which is approximately equal to 1353 ± 21 W/m^2^. In the solar radiation spectrum, ultraviolet (UV) radiation with a wavelength of 100~400 nm provides about 117.7 W/m^2^ of energy, accounting for only 8.7% of the total solar radiation energy, but it has a very important impact on the properties of materials, especially polymer materials. According to the characteristics of photons of different wavelengths, UV radiation can be further subdivided into vacuum ultraviolet radiation with a wavelength of less than 200 nm, medium ultraviolet radiation with a wavelength of 200~320 nm, and near ultraviolet radiation with a wavelength of 320~400 nm [2].

UV irradiation has a very important effect on the properties of materials [3,4,5]. In the space environment, solar ultraviolet irradiation will have a serious impact on the tribological properties of solid lubricants such as polymer self-lubricating coating, MoS_2_ coating and diamond-like film [6,7,8], especially some polymer materials, but these polymer materials have good mechanical properties, high wear resistance, excellent self-lubricating properties and chemical stability, and many other advantages, and are widely used in the aerospace field, so researchers are required to improve these materials, the current research on the impact of space UV irradiation on material properties. There are two main methods: one is the direct space exposure test [9,10,11], the method to get the most accurate results, but obviously this is not easy to achieve, so this is not a conventional test, the second is to simulate the test on the ground, which is currently the best means to study the impact of space environmental factors on materials, and, ground simulation test is usually designed as a kind of accelerated test, simulating the environment on the ground for a week, and the test is the same as in the vehicle in space. is equivalent to tens to hundreds of weeks of exposure of the vehicle in space, which provides an effective guarantee of increased service time of the spacecraft.

Ground simulation tests of space ultraviolet rays were carried out very early, such as the Marshall Space Flight Center of NASA completed the design and put into use of CEETC2 and CEETC3 systems in 1990 and 1996, respectively. Sugimura [12] studied the performance changes of organic silane self-assembled monolayers under vacuum ultraviolet irradiation. The wavelength of vacuum ultraviolet ray used was 172 nm, and the irradiation intensity was 100 W/m^2^, which was about 1 times the solar constant; Tokoroyaman [13,14] was used to test the UV radiation resistance of several carbon films. The equipment used can emit ultraviolet rays at 254, 312 and 365 nm. Can only emit a specific wavelength of ultraviolet rays is not very good simulation of space environment; Zhang [15] designed a set of space UV simulation equipment in 1995. The UV wavelength range is 200–400 nm, the radiation intensity is 200~400 W/m^2^, about three times the solar constant; The UV radiation intensity used in the study of Géraldine Theiler was only 40 W/m^2^ [16]; the space environment simulation test machine developed by SUN can simulate UV rays with wavelength of 115~400 nm and irradiation intensity of 300 W/m^2^ [17,18]. The UV wavelength range of these devices meets the requirements of simulated space environment, but the UV radiation intensity is slightly lower. WU [19] had studied the influence of UV radiation on Ag/a-C:H nanocomposite films. The UV radiation wavelength is 115~400 nm, the irradiation intensity is 600 W/m^2^, about 6 times the solar ultraviolet intensity. The UV intensity of the device has been improved, but the device can not be used for in-situ tribological performance test.

In this study, a new type of simulated space ultraviolet irradiation device was developed through a series of work such as the selection of light source and the design of optical path, so that it can produce ultraviolet rays that are the same as the ultraviolet spectrum of solar light. At the same time, the device can be used in conjunction with the existing MSTS-1 vacuum friction and wear tester [20], which realizes the requirements for the simulation of space ultraviolet irradiation and the in-situ tribological performance test. After the investigation of this type of equipment, the technical indexes shown in Table 1 are formulated, and the operation effect of the device is tested and verified by polytetrafluoroethylene (PTFE) material. PTFE and its composite materials are a kind of solid lubrication materials that are widely used in the current space environment due to their low friction coefficient and good thermal stability. According to Yu et al. at reference [21], the PTFE has the role of lubrication coatings in poly(methyl/butyl methacrylate) composite-based materials. Song et al. in reference [22] presented the PTFE and Al_2_O_3_ filler role in the polyimide composites, to improve the anti-irradiation and wear resistance. In reference [23] Skurat et. al. talked of the multilayer thermo-isolation of the spacecraft with the organic polymer films, including 100–200 micrometer thick of PTFE copolymer. They specify that PTFE copolymer or other polymers are prone to degradation by a number of destructive factors in the space environment like mass losses and mechanical degradation of PTFE during vacuum UV). And in the last reference [24] the effect of gamma (γ) irradiation on the tribological behavior of PTFE coatings under vacuum conditions was also discussed. The PTFE was presented as a popular polymer solid lubricant and some characteristics of it, like resistance to the chemical attack in a wide variety of solvents and solutions, high melting point, low coefficient of friction, and biocompatibility. Depending on the filler, the composite PTFE has better wear resistance. In their experiments, Yuan et al., the PTFE coatings were fabricated on LY12 substrates with a diameter of 70 mm and a depth of 10 mm by PVD (physical vapor deposition) method on both sides. 

## 2. Materials and Methods

### 2.1. Development of a Space Ultraviolet Radiation Simulator

#### 2.1.1. Integral Structure of Space Ultraviolet Radiation Simulator

According to the above design indexes, the developed space UV irradiation device should not only cover the whole UV spectrum range as much as possible, but also realize that the radiation intensity is far greater than that of the sun in the same wavelength range, so as to carry out accelerated test. It is planned to simulate vacuum UV radiation, near UV radiation and medium UV radiation respectively by using double light sources. Among them, short-wavelength vacuum UV rays cannot pass through the quartz window and are easily absorbed by the air, so the deuterium lamp is directly hung in the vacuum chamber and near the sample table for simulation. A special light source and light path transformation mechanism are used to simulate the near UV radiation device. The light emitted by the two light sources is not linear, and the middle superposition area can completely cover the middle ultraviolet spectrum. The structure of the whole set of UV irradiation simulator integrated with MSTS-1 is shown in Figure 1.

L7293-50 deuterium lamp produced by Hamamatsu company of Japan was selected as the light source for simulating vacuum UV. Its shape and luminous spectrum distribution are shown in Figure 2. The deuterium lamp has a power of 30 W and a continuous spectrum of 115~400 nm, with its peak wavelength at 160 nm. The radiation window with a diameter of 15 mm is located on the side wall of the cylindrical deuterium lamp (extended long nose glass tube), and the window material is MgF_2_. As shown in Figure 2, the deuterium lamp is directly installed near the sample table with an incident angle of about 45° and the distance between the light emitting point of the deuterium lamp and the surface of the sample is about 60 mm.

#### 2.1.2. Structure and Working Principle of Near UV Irradiation Device

The near UV irradiation device is mainly composed of light source, power supply and controller. The power supply and control circuit include high power steady current source, timing alarm, programmed electric shutter, high voltage mercury xenon lamp trigger circuit and protection circuit. The structure of the light source device is shown in Figure 3, which is composed of a light source, a focusing mechanism, a concentrator, a UV transmission filter, an integrating lens group and a collimator. The light path principle of the near UV irradiation device is shown in Figure 4. The light source is located at the first focus (F_1_) of the ellipsoid reflector, and the light from the light source converges to the optical integrating mirror group near the second focus (F_2_). The integrating mirror group superposes the light lines in different space regions and projects them to the collimator. The collimator converges the divergent light into parallel beams and parallel beams. It acts directly on the sample table in the vacuum chamber through the quartz window.

The optical path design scheme is helpful to ensure the intensity, uniformity and collimation of ultraviolet irradiation on the sample table. The design process and principle of key components in optical path are introduced as follows:

(1) Choice of light source: the energy conversion efficiency of the UV light source is usually 8~10%. To achieve the irradiation intensity of three UV constants in the range of 50 mm × 50 mm, the power of the light source is about 350 W. The GXZ500 type 500 W spherical high-pressure mercury xenon lamp produced by Shanghai Hualun Bulb Factory was selected as the near UV light source. Its appearance is shown in Figure 5a, and the working current is 20 A. This kind of bulb has the characteristics of high reliability, good stability and fast starting, and its service life is generally more than 2000 h. As shown in Figure 5b, the UV spectrum emitted is between 115 and 404 nm, and the near UV component is strong.

(2) Design of condenser: the function of the condenser is to converge the light emitted from the spherical high-pressure mercury-xenon lamp source to the vicinity of the second focal plane, so as to make full use of the luminous energy of the mercury-xenon lamp to meet the requirements of UV radiation energy and help improve the unevenness of irradiation.

Because of the axial symmetry of the mercury-xenon lamp light source, the use of an ellipsoidal reflective condenser can obtain a better concentrating effect. The overall dimensions of the condenser lens can be calculated by Equations (1) and (2):(1)$$y^{2}{=2R}_{0}{X-(1-e}^{2}{)x}^{2}$$
(2)$$y=f_{2}\mathrm{tgu}-\mathrm{xtgu}$$
where R_0_ and e can be calculated by Equations (3) and (4):(3)$$R_{0}=\frac{{2f}_{1}f_{2}}{f_{1}{+f}_{2}}$$
(4)$$e=\frac{f_{2}{-f}_{1}}{f_{1}{+f}_{2}}$$
where: R_0_ is the vertex curvature radius; e is the eccentricity; f_2_ is the second focal length of the ellipsoidal condenser, f_2_ = M_0_f_1_; f_1_ is the first focal length of the ellipsoidal condenser; M_0_ is the paraxial imaging magnification of the ellipsoidal condenser; u is the aperture angle. Figure 6 shows the relationship between the overall dimensions of the ellipsoidal reflector condenser.

In order to reduce the loss of light energy as far as possible, the ellipsoidal reflector adopts numerical control processing to ensure the accuracy of surface shape, and the surface vacuum deposition of highly reflective metal film has a surface reflectance of more than 95%.

(3) Ultraviolet transmittance optical plate: in order to eliminate the non-ultraviolet components in the spectrum emitted by the mercury xenon lamp light source, the light is filtered before entering the lens group of the integrator field. The ultraviolet filter is made of quartz glass substrate and coated with high temperature dielectric film in vacuum. The transmittance of UV light from 200 to 400 nm is more than 90%, and the transmittance of visible light and infrared light is less than 10%.

(4) Optical integrating mirror group: the function of the optical integrating mirror group is to project the light radiation of different space areas onto the collimator. The integrator is the most important optical element to realize the uniform distribution of irradiance in the irradiation plane. The integrator lens group is divided into two groups: the field lens group and the projection lens group. Each group is composed of multiple small element lenses and the supporting base lens. All element lenses and supporting lenses of the integrated mirror group are made of UV transparent quartz glass, and then connected by optical glue technology to ensure the stability of long-term work under high temperature conditions. According to the design index, it is necessary to ensure that the uniform irradiation area is not less than Φ50 mm. After preliminary calculations, the overall dimension of the element lens should be designed as a regular hexagon less than 10 mm for splicing, but it is very difficult to process such a size element lens into hexagon, which not only greatly increases the cost, but also can’t guarantee the processing accuracy, which affects the realization of the overall index of the equipment. For this reason, the shape of the element mirror is changed to square, and the shape of the corresponding irradiation surface is also changed to square. It is finally determined that the field lens and projection lens of the integrating lens group are arranged closely with 9 square element mirrors with a side length of 10 mm and a center thickness of 2 mm. The size of the photoresist base is Φ 60 mm × 8 mm. The structure of the optical integrating mirror group is shown in Figure 7.

(5) The design of collimating mirrors: the function of the collimator is to gather the light from the integrator into a parallel beam. As shown in Figure 8, the modified cone axis ellipsoid is selected as the collimating mirror type in this design, which can effectively improve the radiation nonuniformity of the second focal plane while keeping the original imaging relationship unchanged. The detailed dimensions are as follows: the outer diameter is Φ 96 mm, the front opening is Φ 86 mm, the clear aperture is Φ 74 mm, the central cone angle is 1.5° and the thickness is 16 mm.

In this design, the collimator lens is installed in the flange of the rubber ring, which not only plays the role of converging the beam, but also serves as the sealed quartz window of the vacuum chamber. This design can reduce the attenuation of UV light by a quartz window. The collimator is made of quartz optical glass. In order to minimize the loss of UV light energy, the ultraviolet antireflective film is deposited on the surface of collimator in vacuum.

(6) Focusing mechanism: The focusing mechanism adopts a manual mechanical 3D combined displacement adjustment platform, which is used to adjust the arc position of the spherical high-pressure mercury lamp so that it is at the first focus of the ellipsoidal reflection condenser.

### 2.2. Test Verification Method

#### 2.2.1. Material

The PTFE sample material used was cut from the PTFE die casting plate with a thickness of 6 mm. The raw material was prepared by Beijing Plastics Factory, and the purity of raw materials was about 99.99%. A group of PTFE samples were irradiated by the UV simulation device for 10 h in vacuum environment.

#### 2.2.2. Experimental Approach

The developed UV irradiation simulation device was installed on the self-developed MSTS-1 vacuum friction and wear tester, and the ball-on-disc friction and wear test of PTFE samples was carried out in vacuum environment. The other group of samples were directly subjected to vacuum friction and wear test without UV irradiation. When the load was 3 N, the speed was 100 RPM, the grinding material was GCr15, the ball diameter was 9.525 mm, and the test time was 1200 s. 

The wear scar morphology was characterized by scanning electron microscope (SEM). The equipment model was Navo NanoSEM 450 field emission scanning electron microscope. The optical design of near-UV irradiation device was verified by virtual operation test using LightTools optical modeling software. Then, The three-dimensional surface morphology of PTFE samples before and after UV irradiation was observed by Nanoscope V atomic force microscope (AFM). The scanning area was 40 × 40 μm. The phase composition and chemical composition changes of PTFE sample surface before and after UV irradiation were analyzed by X-ray photoelectron spectroscopy (XPS). The spectrometer was ESCALab220i XLX-ray photoelectron spectrometer produced by Semerfeld. It used Mg-Kα excitation source with power of about 300 W and spot diameter of about 500 μm.

A self-developed indentation device was used to test the mechanical properties of PTFE samples such as fracture toughness and plastic deformation resistance before and after UV. The device was equipped with an acoustic emission detector [25,26]. When the material breaks or cracks occur, the instantaneous stress waves are released, which can be used to continuously detect the evolution of slight deformation and internal damage [27,28]. Acoustic emission detectors are very sensitive to these stress waves and can be converted to acoustic emission signals that are easy to read and analyze [29,30]. By analyzing the typical parameters of acoustic emission signals, some mechanical properties of samples can be indirectly characterized.

## 3. Results and Discussion

### 3.1. Operation Effect Test of Space Ultraviolet Radiation Simulator

#### 3.1.1. Parameter Test and Calibration of the Whole Space Ultraviolet Radiation Simulation Device

Under the conditions of deuterium lamp working current of 300 mA and mercury xenon lamp current of 20 A, according to the method specified in GB/T12637-90 ‘General specification of solar simulator’, the key parameters of the whole set of UV irradiation device are tested according to the above technical indicators. 

(1) Measurement of irradiance: the ultraviolet irradiance intensity was measured by the ultraviolet irradiance meter verified by the Chinese Academy of Metrology. The results show that the average irradiance is 590 W/m^2^ in the range of Φ 50 mm, which is about 6 solar constants.

(2) Measurement of irradiation non-uniformity: Irradiation non-uniformity refers to the difference of irradiance in different areas in the irradiation space. Generally, it can be divided into surface irradiation nonuniformity and volume irradiation nonuniformity, which can be calculated according to Equation (5):(5)$$\frac{∆E}{E}=\frac{E_{\max}{-E}_{\min}}{E_{\max}{+E}_{\min}}\times 100\%$$
where: ∆E/E is the nonuniformity of irradiance; E_max_ is the maximum irradiance on the irradiation surface (or within the volume); E_min_ is the minimum irradiance; E is the average irradiance. The irradiance intensity at the center and four vertices of the square spot was measured by UV irradiance meter. The maximum and minimum irradiance in the irradiation plane were 630 W/m^2^ and 565 W/m^2^ respectively. According to Formula (5), the irradiance nonuniformity was ±5.44%.

(3) Measurement of quasi-right angle: using the pinhole imaging method, the spot size at different irradiation distances of the near-UV irradiation device was measured, and the quasi right angle θ = 4 of the near-UV irradiation device was calculated by Equation (6):(6)$$\theta =\pm \mathrm{tg}\frac{D}{2H}$$
where: D is the diameter of the image; H is the distance from the small hole to the imaged surface.

(4) Measurement of irradiation instability: irradiation instability refers to the change in irradiance over time during long-term operation of the UV irradiation device. The irradiation instability can be obtained from Formula (7):(7)$$\frac{∆E}{{E|}_{T}}=\frac{E_{\max}{-E}_{\min}}{E_{\max}{+E}_{\min}}{\times 100\%|}_{T}$$
where: E_max_, E_min_ is the maximum and minimum irradiance in the measurement time T; ∆E/E|_T_ is the irradiation instability in time T. Using constant temperature silicon photocell as detection element, the irradiance of the sample center is measured every 1 h during the continuous operation of the UV irradiation device for 15 h. The calculation shows that the irradiation instability of the developed UV irradiation device is (1%)/h.

To sum up, the comparison of design index and actual index of UV irradiation device is shown in Table 2. It can be seen that the self-designed UV radiation simulation device can meet all the requirements of space UV environment simulation, and can be used for subsequent tests.

#### 3.1.2. Virtual Operation Effect of Near UV Irradiation Device

The above optical design of the near UV irradiation device was tested by LightTools optical modeling software. The simulation light path is shown in Figure 9, which is mainly composed of condenser, integrator and collimator. Figure 10 shows the simulation model and operating effect of the near UV irradiation device. It can be seen that the spot area is about 60 mm × 60 mm at the irradiation distance of 190 mm, and the irradiation intensity is very uniform within the range of 50 mm × 50 mm.

### 3.2. Experimental Validation

#### 3.2.1. Surface Morphology and Microscopic Analysis

Figure 11 shows the color change on the sample surface after 10 h of UV irradiation. It can be seen that the original sample is milky white. After UV irradiation, the sample surface color deepened to brown. The reason of color change may be: after UV irradiation, high-energy UV photons act on the surface of the sample, resulting in part of the molecular chain fracture, while triggering some bond recombination, the sample surface ‘carbonized’, thus deepening the color [31].

The three-dimensional surface morphology of the samples before and after UV irradiation is shown in Figure 12. The surface of the original sample is overall smooth, but there are large protrusions and pits in some places (Figure 12a). After UV irradiation (Figure 12b), the surface of the sample was smoother, and the values of the average roughness Ra and the root mean square Rq of the sample showed a decreasing trend. This rule is consistent with the previous research results [32,33,34], indicating that the UV irradiation device described in this paper can effectively carry out the related research work on the influence of ground simulated space UV irradiation on the surface properties of materials.

#### 3.2.2. XPS Analysis

The XPS C1s spectra of PTFE samples before and after UV irradiation are shown in Figure 13. The C1s peaks were fitted. The C1s spectra of unirradiated samples were fitted to four components at 292.29, 291.58, 284.79 and 284.17 eV, corresponding to C-F_3_, C-F_2_, C-O and C-C bonds, respectively. However, the bond structure of the samples changed greatly after irradiation. For the samples after UV irradiation, it can be found that the peak intensities of C-F_3_ and C-F_2_ bonds are significantly reduced, while the peak intensities of C-O and C-C bonds are significantly increased. In addition, the fifth peak at 288.83 eV corresponds to the C=O bond, which is mainly caused by the oxidation of carbon on the surface of the sample. Figure 13a,c are full spectrum of XPS. The changes of element content on the surface of PTFE samples were summarized in Table 3. It can be seen that the content of F element on the surface of PTFE samples decreased greatly after UV irradiation, and the content of C and O elements increased greatly. The photon energy of vacuum UV is up to several or even dozens of electron volts, which is larger than the bond energy of chemical bonds [35,36]. UV irradiation may destroy the C-F bond in the surface molecules of the sample, and oxidize the surface of the sample, and accumulate a lot of carbon on the surface.

#### 3.2.3. Mechanical Properties Analysis

Figure 14 shows the amplitude variation curve of acoustic emission signal during indentation. It can be seen that in the stable period, the signal amplitude before UV irradiation is maintained at about 30.2 dB, and the signal amplitude after UV irradiation is maintained at about 28 dB. But for samples before UV irradiation, the signal vibration is low, which means that the mechanical properties of samples before UV irradiation are more uniform. In the first 20 s loaded into the sample, the acoustic emission signal amplitude of the sample before UV irradiation is low, indicating that the sample before UV irradiation has better ductility and toughness than that after irradiation. Based on the above results, it can be concluded that long-term UV exposure experiments under this equipment will have a great impact on the mechanical properties of the material.

#### 3.2.4. Tribological Performance Analysis

Figure 15 compares the tribological properties of the samples before and after UV irradiation. It can be seen from the figure that before UV irradiation, the sample is initially in a very smooth state. As the wear progresses, the friction coefficient is increasing. At about 70 s, the friction coefficient gradually tends to be stable. At this time, the variation range of the friction coefficient is very small. After about 150 s, the friction coefficient amplitude of the sample surface increases suddenly due to wear and fluctuates around 0.12. And after 1200 s, the friction coefficient began to fluctuate around 0.14. After UV irradiation, the friction coefficient of the sample surface fluctuates around 0.10 in the stable period. After 1050 s, the friction coefficient of the sample surface fluctuates around 0.14 after UV irradiation. This means that UV irradiation significantly increases the brittleness of the surface layer of the sample, so the fluctuation range of the friction coefficient increases. However, the effect of UV on the sample is limited to the surface layer of the sample. After a period of friction scraping, the “brittle layer” on the surface is removed, so the average value of the friction coefficient is not significantly different from that of the unirradiated sample.

Figure 16 shows the comparison of the wear scar morphology of the sample after 10 h of UV exposure. It can be seen from Figure 16 that before UV irradiation, the surface of the sample is relatively smooth, with only a few light-colored peeling pits randomly distributed. After UV irradiation, the surface of the sample peeled off seriously, and the peeling pit was deeper, and the color was aggravated. After 10 h of UV irradiation, the surface of the sample became relatively flat, with clearly visible wear marks, which were distributed in the same direction and arranged neatly. After the irradiation, the arithmetic average roughness and root mean square roughness of the sample surface decreased to a certain extent, indicating that the sample exhibited good tribological properties after UV irradiation.

## 4. Conclusions

(a)In order to study the tribological properties of lubricating materials under space UV irradiation environment, an in-situ Tribological Testing Machine for simulating space UV radiation on ground conditions was developed. The relevant indexes of the device were as follows: the spectral range was 115~404 nm, average irradiation intensity is 590 W/m^2^, and the uniform radiation area was 50 mm × 50 mm. Therefore, the device can not only emit high-intensity space ultraviolet rays, but also meet the requirements of in-situ friction performance test.(b)Through the UV irradiation test of PTFE, which is commonly used as solid lubricating material, the test machine can induce the damage and failure of PTFE material similar to space flight test. After irradiation with high-energy UV photons, some molecular chains on the surface of the sample are broken, and some bonds are recombined. The surface of the sample is carbonized, the color is deepened, and the surface roughness of the sample is reduced.(c)After UV irradiation, the tribological and mechanical properties of PTFE materials will change. The brittleness of the surface layer of PTFE increases obviously, and the fluctuation range of friction coefficient becomes larger in the process of friction and wear test. However, the action depth of UV irradiation is limited to the surface layer of the sample. After removing the “brittle layer” on the surface of the sample for a period of time, the average value of the friction coefficient during the stable period is not significantly different from that of the unirradiated sample.

## Figures and Tables

**Figure 1 materials-15-02063-f001:**
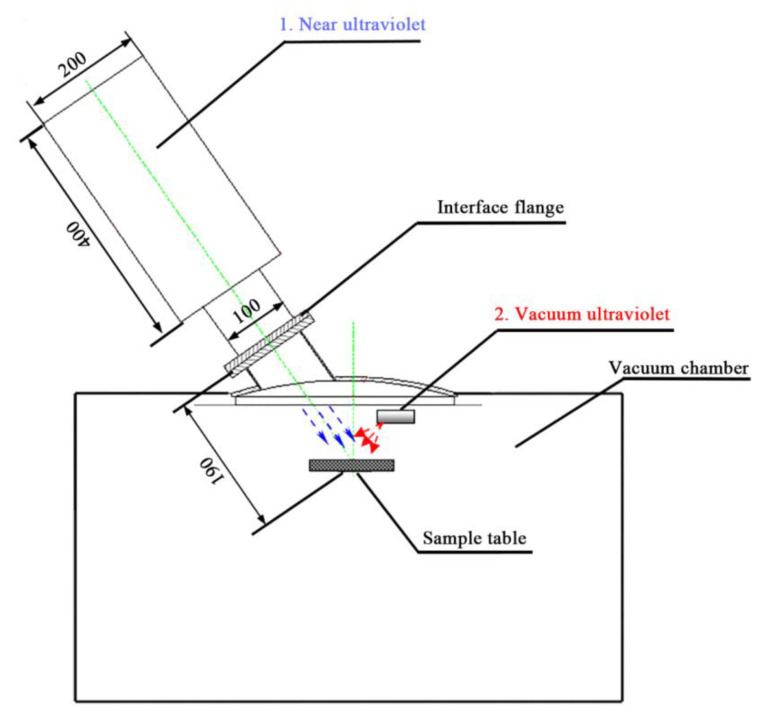
Structure diagram of space ultraviolet irradiation device.

**Figure 2 materials-15-02063-f002:**
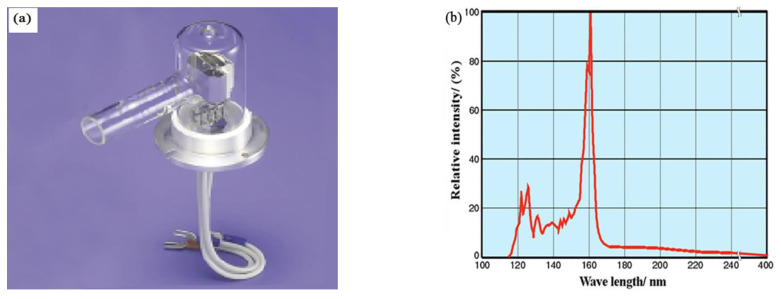
(**a**) Appearance of L7293-50 deuterium lamp; (**b**) spectrum distribution curve.

**Figure 3 materials-15-02063-f003:**
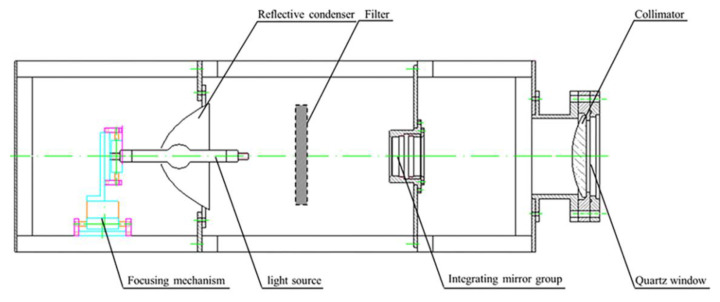
Schematic diagram of near ultraviolet light source device.

**Figure 4 materials-15-02063-f004:**
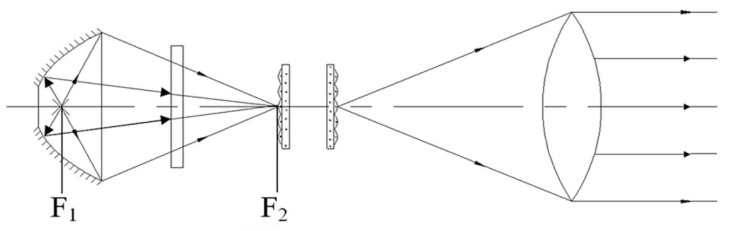
Schematic diagram of light path principle.

**Figure 5 materials-15-02063-f005:**
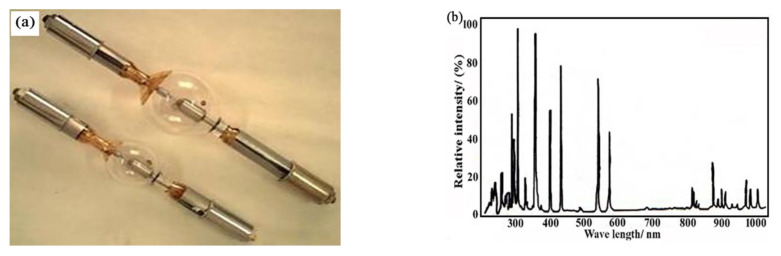
(**a**) Appearance of gxz500 mercury xenon lamp; (**b**) Luminescence spectrum.

**Figure 6 materials-15-02063-f006:**
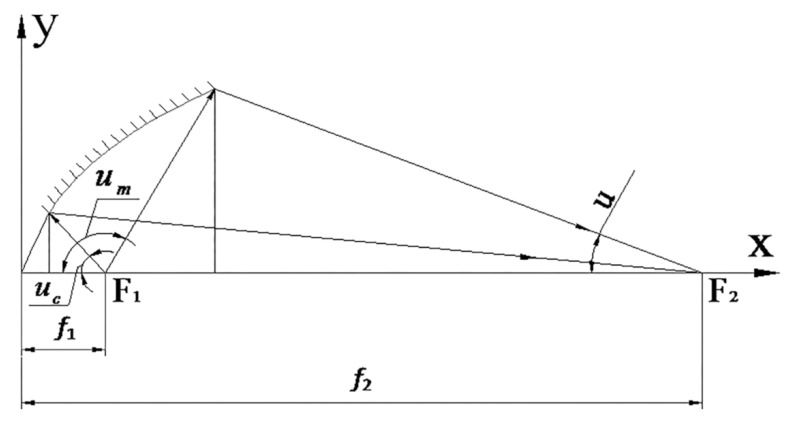
Dimensional relationship of ellipsoidal reflective condenser.

**Figure 7 materials-15-02063-f007:**
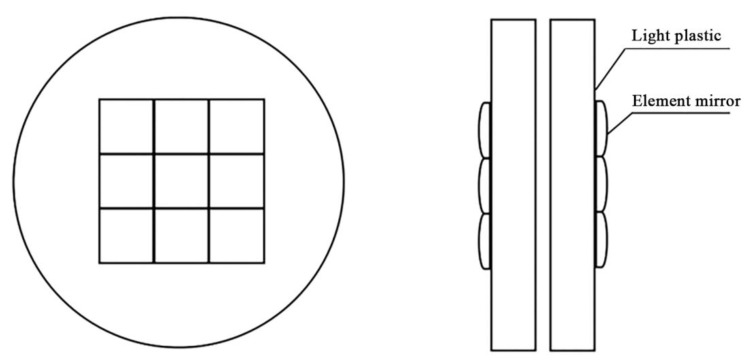
Schematic diagram of the structure of the integrating mirror group.

**Figure 8 materials-15-02063-f008:**
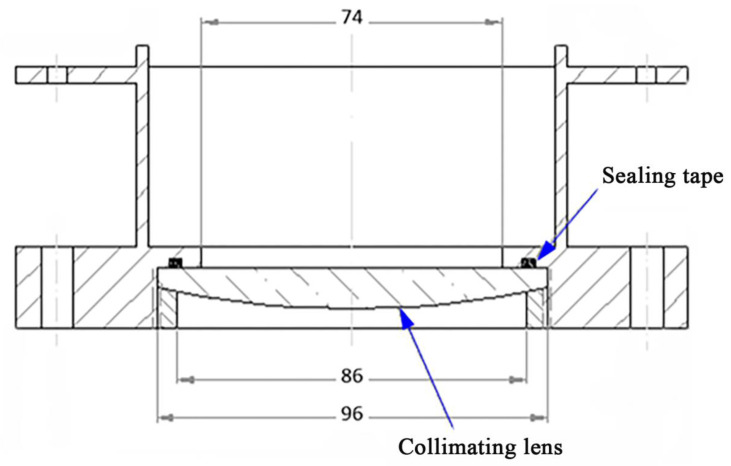
Structure and size of collimator.

**Figure 9 materials-15-02063-f009:**
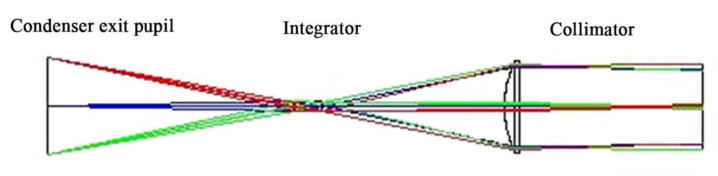
Analog optical path layout diagram.

**Figure 10 materials-15-02063-f010:**
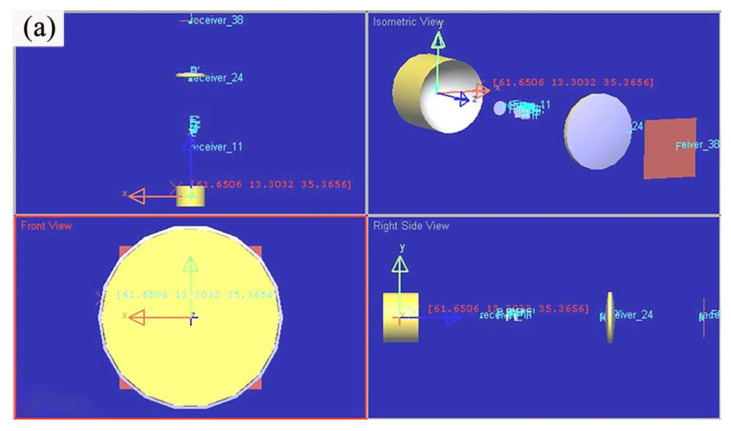
(**a**) LightTools software simulation model of near ultraviolet irradiation device (**b**) irradiation effect.

**Figure 11 materials-15-02063-f011:**
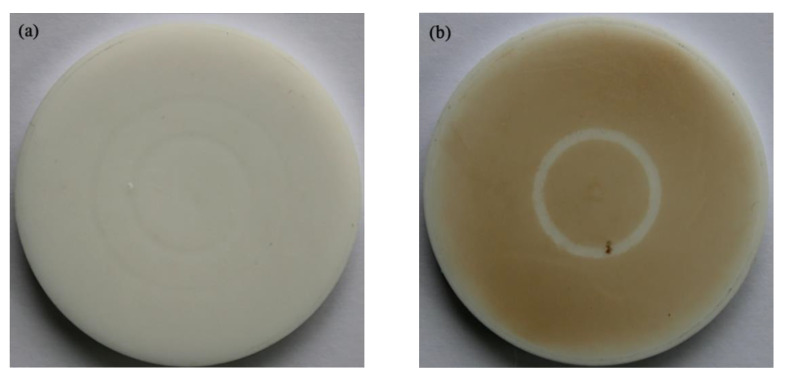
Appearance photos of sample samples before and after UV irradiation: (**a**) Before irradiation; (**b**) After irradiation.

**Figure 12 materials-15-02063-f012:**
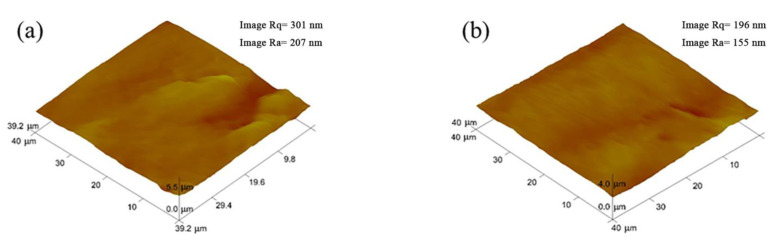
The surface topography of the sample before and after UV irradiation: (**a**) Before irradiation; (**b**) After irradiation.

**Figure 13 materials-15-02063-f013:**
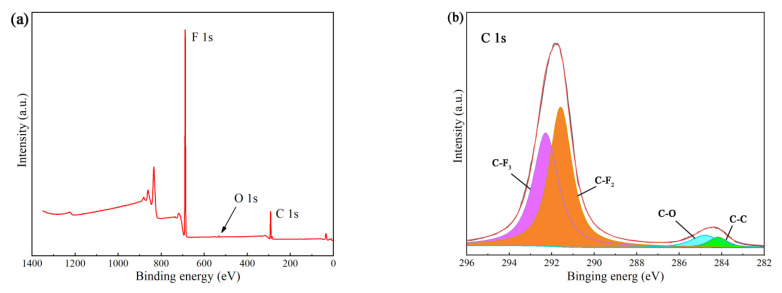
XPS analysis of samples before and after UV irradiation: (**a**) full spectrum before UV irradiation; (**b**) C 1s fine spectrum before UV irradiation; (**c**) full spectrum after irradiation; (**d**) C 1s fine spectrum after irradiation.

**Figure 14 materials-15-02063-f014:**
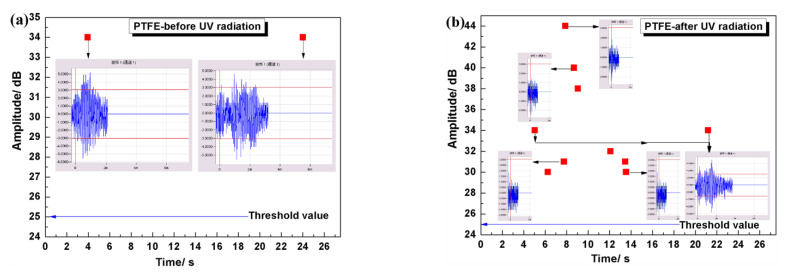
The change curve of the amplitude of the acoustic emission signal before and after UV irradiation of the sample during the indentation process: (**a**) After irradiation; (**b**) Before irradiation.

**Figure 15 materials-15-02063-f015:**
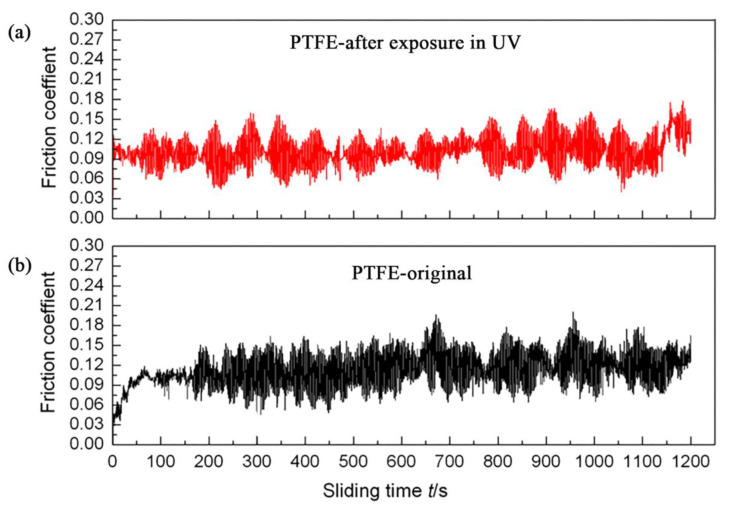
Comparison of the change trend of the friction coefficient of the sample surface before and after UV irradiation: (**a**) After irradiation; (**b**) Before irradiation.

**Figure 16 materials-15-02063-f016:**
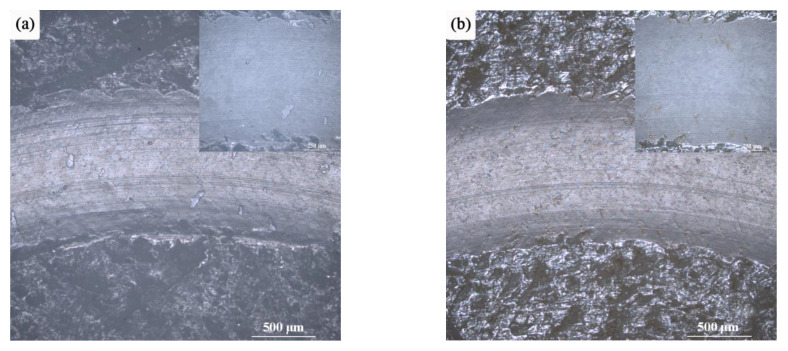
Comparison of the wear scar morphology of the sample before and after UV irradiation: (**a**) Before irradiation; (**b**) After irradiation.

**Table 1 materials-15-02063-t001:** Technical indexes of space ultraviolet irradiation simulator.

UV Irradiation Intensity	Spectral Range	Uniform Irradiation Area	Quasi-Right Angle	UV Irradiation Uniformity	UV Irradiation Stability
5 UV constants	115~404 nm	Φ50 mm	Φ50 mm	better than ±10%	better than (5%)/h

**Table 2 materials-15-02063-t002:** Comparison of design index and measured index of UV irradiation simulation device.

	Design Index	Measured Index
Spectral range	120~400 nm	115~404 nm
Irradiation intensity	3 UV constants	6 UV constants
Irradiation surface size	≥Φ50 mm	50 mm × 50 mm
Irradiation inhomogeneity	Better than ±10%	±5.4%
Quasi right angle	Less than 5°	4°
Irradiation instability	Better than (5%)/h	(1%)/h

**Table 3 materials-15-02063-t003:** Changes of element content on PTFE surface before and after UV irradiation.

Element	Before Irradiation (at %)	After Irradiation (at %)
F	88.87	45.04
C	10.50	36.96
O	0.63	18.00

## Data Availability

Not applicable.

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
