# Peer review of "Development and Verification Experiment of In-Situ Friction Experiment Device for Simulating UV Irradiation in Space"

_materials, 2022, doi:10.3390/ma15062063_

Round 1

Reviewer 1 Report

The authors of the paper”  Development and verification experiment of in-situ friction experiment device for simulating UV irradiation in space”,   took into consideration the previous suggestions of reviewers for improving the quality of the manuscript,  including, for instance, in the introduction (i) the effect of UV irradiation on the performance (good mechanical properties, high wear resistance, excellent self-lubricating properties, and chemical stability) of some polymer materials, making suitable for the aerospace applications and (ii) provided sufficient background and included relevant references, etc.

 In the experimental part, the authors developed the test verification method. 

However, previous to the publication, the work still needs some improvement, such as:

1) In section 2.2.  as the last request to develop more about the  characteristics of  the material (PTFE) used, you only specify, quote” PTFE and its composite materials are a kind of solid lubrication materials that are widely used in the current space environment due to their low  friction coefficient and good thermal stability [15-18].”

As a suggestion, the authors can improve the data regarding the PTFE, like:

According to Yu et al. at reference [15], the PTFE has the role of lubrication coatings in poly(methyl/butyl methacrylate) composite-based materials.

Song et. al. in reference [16] presented the PTFE  and Al2O3 filler role in the polyimide composites, to improve the anti-irradiation and wear resistance.

In reference [17] Skurat et. al. talked of the multilayer thermo-isolation of the spacecraft with the organic polymer films, including 100-200 micrometer thick of PTFE copolymer. They specify that PTFE copolymer or other polymers are prone to degradation by a number of destructive factors in the space environment like mass losses and mechanical degradation of PTFE during vacuum UV).

And in the last reference [18] the effect of gamma (γ) irradiation on the tribological behavior of PTFE coatings under vacuum conditions was also discussed. The PTFE was presented as a popular polymer solid lubricant and some characteristics of it, like resistance to the chemical attack in a wide variety of solvents and solutions, high melting point, low coefficient of friction, and biocompatibility. Depending on the filler, the composite PTFE has better wear resistance. In their experiments, Yuan et. al., the PTFE coatings were fabricated on LY12 substrates with a diameter of 70 mm and a depth of 10 mm by PVD (physical vapor deposition) method on both sides.

2) Lines 224 and 225 specified:” the three-dimensional surface morphology of PTFE samples before and after UV irradiation was observed by atomic force microscope (AFM)”. Please provide the methodology of testing by AFM.

3) Lines 226 and 227 specified „The phase composition and chemical composition changes of PTFE sample surface before and after UV irradiation were analyzed by X-ray photoelectron spectroscopy (XPS)”. Please provide the methodology of analysis by XPS.

4)Please reorganize the structure of experimental parts like:

At  „2.2. Test verification method” insert subsection like:

 Lines 240-243:

 2.2.1 Material. „The PTFE sample material used was cut from the PTFE die casting plate with a thickness of 6 mm. The raw material was prepared by Beijing Plastics Factory, and the purity of raw materials was about 99.99 %. A group of PTFE samples was irradiated by the UV simulation device for 10 h in a vacuum environment”

Lines 243-248:

2.2.2. Experimental approach. „The ball-on-disc friction and wear test was carried out in a vacuum environment. The other group of samples was directly subjected to vacuum friction and wear test without UV irradiation. When the load is 3 N, the speed is 100 r/min, the grinding material is GCr15, the ball diameter is 9.525 mm, and the test time is 1200 s.

The wear scar morphology is characterized by a scanning electron microscope (SEM).

5) Line 246: Use RPM instead of  “r/min.”,

6) Line 248: Please provide the methodology of testing by SEM.

7) Line 65: Please check Hiroyuki [6]. Doesn’t exist.

8)  I didn’t find the following references [11], [20] Can you give us a DOI?

  [11].  Sun, X. J.; Liu, W. M. Tribological testing technology in simulated space environments. Engineering & Test. 2009, S1, 24-29.

[20]. Wang, H. D.; Xu, B. S.; Song, Y. N.; Piao, Z. Y. Coating bonding strength tester for remanufactured parts, CN103196824A, 10 464 July, 2013

9) Please verify the reference [21] (https://doi.org/10.4028/www.scientific.net/ssp.267.253 ) and insert at the correct number of volume, for instance: Song, J. F., Zhao, G., Ding, Q. J., & Qiu, J. H. (2017). Anti-irradiation and wear resistance of polyimide composites. In Solid State Phenomena (Vol. 267, pp. 253-257). Trans Tech Publications Ltd.

Reviewer 2 Report

Dear Authors,

The manuscript is slightly modified. However, the conclusion below is still confusing:

  (b) “After irradiation with high-energy  UV  photons, some molecular chains on the surface of the sample are broken, and some bonds are recombined.”

Please explain how the recombination phenomenon was investigated and determined? Was it determined by XPS analysis? What is the scale of this phenomenon? Can you provide quantitative results or photos at high magnification?

Round 2

Reviewer 2 Report

Dear Authors,

Since the doubts were cleared, I have no more comments.

This manuscript is a resubmission of an earlier submission. The following is a list of the peer review reports and author responses from that submission.

Round 1

Reviewer 1 Report

Dear Authors,

The subject is interesting in general. However, the manuscript is rather a technical report focused on description of the design and construction of the devices. It does not meet the high requirements for publication in the Materials.

Abstarct

„...it is found that the in-situ UV irradiation device can achieve effect of irradiation... the irradiation will lead to changes in the structure and performance polymer material surface.”

Comment: Both above finding seems to be trivial.

Introduction

This section is too short and unsatisfactory. It provides just basic information on solar radiation in the space. There is no discussion based on referenced literature, no state of the art.

Results and discussion

“After  UV  irradiation, high-energy  ultraviolet  photons  act  on  the  surface  of  the  sample,  causing  part  of  the molecular chain to be broken, and at the same time inducing some bond recombination, the surface of the sample is "carbonized" and the color deepens [6].”

Comment: Since it is not based on the results of your study, the above part is a part of literature review.

“The  surface  morphology  of the  sample  is  an  important  factor that determines the appearance and performance of the sample   [7].”

Comment: The above is another example of literature review. Moreover, the Figure 12 does not correspond or prove the statement.

“In order to explore the  corrosion mechanism of UV radiation on the  sample...”

Comment: What do you mean “corrosion mechanism”?

Conclusions

(b) “After irradiation with high-energy ultraviolet photons, some molecular chains on the surface of the sample are broken, and some  bonds are recombined. The surface of the  sample  is carbonized, the  color is deepened, and the  surface roughness of the sample is reduced.”

Comment: The above is not supported by results of your study.

Reviewer 2 Report

The authors fabricated a UV irradiation device that can simulate solar spectrum. However, I am not sure whether this kind of device is necessary to investigate the UV irradiation effect on polymeric materials. This is because, 1) the UV irradiation effect on polymer is well known, and 2) the artificial UV source cannot completely correspond with the solar spectrum. If the purpose of the research is to investigate the effect of UV light in solar spectrum on polymer, there might be no reason why such UV source is required. For example, electrons and protons are also major radiations coming from the Sun and both particles have broad energy spectra. However, nobody tries to use the same energy spectra of the particles to investigate the effects of the two particle radiations on space materials.

I do not thick that the studies are deserved to be published in this journal even though they made big efforts. In addition, the authors can conside the following comments to improve the quality of the manuscript.

  1. Instead of using (Or, in addition to using) vacuum ultraviolet radiation, near ultraviolet radiation and medium ultraviolet radiation, the author can use UVA, UVB, and UVC.

  1. The author considered UV wavelength from 115~404 nm. What is the real UV spectrum of the solar light ? The authors used two UV lamps, and does the UV spectrum achieved using the two lamps correspond with the spectrum of solar light ?

  1. Figure numbering in Page 3 is wrong. For example, 1) As shown in Figure 2, -> As shown in Figure 1. In addition, the Figure caption is wrong. For example, Figure 2 caption should be the spectrum of deuterium lamp, not xenon lamp.

  1. Is there any special reason why the authors chose PTFE as a test polymer material ? PTFE is well known as a radiation-degrading polymer. Space radiation includes energetic proton, enrgetic electrons as well as UV. Therefore, PTFE is not a regid material that can be used in space.

  1. The authors wrote “ After UV irradiation (Figure 12 (b)), the surface morphology of the sample changes greatly.”. They had to describe clearly what changed.

  1. The XPS analysis seems wrong. First,“ the binding energies are 292.5 eV and 284.5 eV, corresponding to CF2 and CH2, respectively”. PTFE does not have CH2 bonds. What is the origin of CH2 ? Second, “The energy is 284.5 eV and 292 293eV, corresponding to CF2 and CH2, respectively.”. This is wrong, too. Probably, 284.5 eV should be attributed to sp3 carbon. In addition, if the authors carefully calibrate the peak position, 284.5 eV should be near 285 eV.

  1. The following sentence “Studies have shown that UV irradiation may break the C-F bonds in the molecules” seems to be wrong. What UV and other irradiation affects is breakage of C-C backbone bonds, which is called “scission or degradation”. PTFE is well-known as a radiation-degrading polymer. In additon, at least one reference should be included in their claims if “studeis have really showed”.

  1. As the authors mentioned, the thickness of the polymer that UV affects is small and the friction coefficient is determined only by the near-surface of the polymer. As a result, the effect of UV on the friction is very limited because the UV-affected surface region is easily peeled off. Here, what is the thickness of the polymer that UV affects ? The authors can determine this by measuring depth-dependent chemical composition of the polymer after the irradiation.

Reviewer 3 Report

In the present paper, the authors investigated the influence of space UV irradiation on spacecraft lubrication materials (PTFE), developed a space UV radiation simulator (set of in-situ friction experimental apparatus), and compared the theoretical, simulated data (i.e. design index)  with the experimental one (the measured index).

The structure and working principle of the in-situ friction experiment device, the parameter test, and the calibration of the whole device are presented in detail. They simulated the vacuum ultraviolet radiation (near-ultraviolet radiation and medium ultraviolet radiation respectively)  by using double light sources. The irradiated PTFE sample was analyzed from a structural and tribological point of view, and indirectly, by analyzing the typical parameters of the acoustic emission signal, some mechanical properties of the sample were analyzed. The morphology of the structure was highlighted by atomic force microscopy and XPS analysis. The authors described also in detail the structure diagram and working principle of near-ultraviolet irradiation device.

The motivation for performing these experiments is from the economic point of view, such as, compared with UV irradiation test in space, ground simulation test has the advantages of convenient operation and low cost. The disadvantage of this approach is the difficulty to obtain an ideal UV beam with approximate concentration and energy similar to the low-orbit environment in space.

The authors concluded that, after stimulation of the conditions similar to the space flight test of the PTFE material, after irradiation with high-energy ultraviolet photons of the samples material, some molecular chains on the surface of the sample were broken, and some bonds are recombined. The surface of the sample is carbonized, the color is deepened, and the surface roughness of the sample is reduced. Also, after UV irradiation, the brittleness of the surface layer of PTFE increases.

The paper, as a whole, is of interest to the scientific community in the material science field, especially due to its specific application.  As a result, I agree with the publication of this paper, after minor revision.  Previous to the publication, I think the following details should be corrected:

  1. Before the first abbreviation that appears in the text please, describe it: for instance: X-ray photoelectron spectroscopy (XPS), polytetrafluoroethylene(???) (PTFE), etc.
  2. In section 2.2. please, put more characteristics regarding the material (PTFE) used.
  3. Please provide the methodology for testing the material from a tribological point of view.
  4. What is the connection between the parameters of the acoustic emission signal, and the mechanical properties of the sample?
  5. Line 352: In the text, you specify that you used a special indentation device. Please give more details about it. The reference [20] is missing.

  1. Please make more clear (improve the image resolutions) of Figure 10(b).

Line 134:  Instead of "formula 1,2 ", put equations (1) and (2).

Line 135: The same, Instead of "formula 3,4" put equations (3) and (4) according to the MDPI template.

Before line 136, put the correct numbers of equations  (3) and (4).

Line 272 „The surface morphology of the sample before and after UV irradiation was observed by atomic force microscope (AFM)” is more suitable to put it on the methodology section.

  1. Please verify and correct the references:

 [3] Ma, G.; Xu, B.; Wang, H.. (2012 ). Development of A Novel Multifunctional Vacuum Tribometer. Chin. J. Vac. Sci. Technol., 32(12), 1-6.

 [4]  Liu, Hai; Hongbin, Geng; Yang, Dezhuang; Abramov, V. V.; Wang, Huaiyi  (2005). Effects of Space Environment Factors on Optical Materials. Journal of Spacecraft and Rockets, 42(6), 1066–1069.         doi:10.2514/1.20885